# Myocardial Hypertrophy and Compensatory Increase in Systolic Function in a Mouse Model of Oxidative Stress

**DOI:** 10.3390/ijms22042039

**Published:** 2021-02-18

**Authors:** Rohan Varshney, Rojina Ranjit, Ying Ann Chiao, Michael Kinter, Bumsoo Ahn

**Affiliations:** 1Aging & Metabolism Research Program, Oklahoma Medical Research Foundation, Oklahoma City, OK 73103, USA; rohanvarshney@gmail.com (R.V.); Rojina-Ranjit@omrf.org (R.R.); ann-chiao@omrf.org (Y.A.C.); mike-kinter@omrf.org (M.K.); 2Cardiovascular Biology Research Program, Oklahoma Medical Research Foundation, Oklahoma City, OK 73104, USA; 3Harold Hamm Diabetes Center, University of Oklahoma Health Science Center, Oklahoma City, OK 73104, USA

**Keywords:** myocardial hypertrophy, CuZnSOD, oxidative stress, reactive oxygen species, systolic function, troponin I, vinculin

## Abstract

Free radicals, or reactive oxygen species, have been implicated as one of the primary causes of myocardial pathologies elicited by chronic diseases and age. The imbalance between pro-oxidants and antioxidants, termed “oxidative stress”, involves several pathological changes in mouse hearts, including hypertrophy and cardiac dysfunction. However, the molecular mechanisms and adaptations of the hearts in mice lacking cytoplasmic superoxide dismutase (Sod1KO) have not been investigated. We used echocardiography to characterize cardiac function and morphology in vivo. Protein expression and enzyme activity of Sod1KO were confirmed by targeted mass spectrometry and activity gel. The heart weights of the Sod1KO mice were significantly increased compared with their wildtype peers. The increase in heart weights was accompanied by concentric hypertrophy, posterior wall thickness of the left ventricles (LV), and reduced LV volume. Activated downstream pathways in Sod1KO hearts included serine–threonine kinase and ribosomal protein synthesis. Notably, the reduction in LV volume was compensated by enhanced systolic function, measured by increased ejection fraction and fractional shortening. A regulatory sarcomeric protein, troponin I, was hyper-phosphorylated in Sod1KO, while the vinculin protein was upregulated. In summary, mice lacking cytoplasmic superoxide dismutase were associated with an increase in heart weights and concentric hypertrophy, exhibiting a pathological adaptation of the hearts to oxidative stress.

## 1. Introduction

Cardiovascular diseases are the leading cause of death in the United States, with a mounting economic burden for medical care [1], and they also compromise exercise tolerance and the life quality of the disease survivors [2,3]. Numerous physiological and pathological conditions lead to complications in cardiac function and morphology, including heart failure, cardiac ischemia–reperfusion, pulmonary diseases, aging, inactivity, chronic inflammation, and diabetes [4,5,6]. Many of these conditions involve multiple alterations in the heart and the whole body (i.e., inflammation), which makes it challenging to isolate one risk factor (i.e., reactive oxygen species) and investigate its contribution to cardiomyopathies in diseases [4]. 

Excess reactive oxygen species (ROS) are one of the common consequences in chronic heart diseases (i.e., chronic heart failure) and physiological conditions (i.e., aging) that lead to cardiac abnormalities. Because of the associations between excess ROS and cardiac diseases, there has been a growing interest in elucidating the mechanisms that regulate cardiac ROS production and detoxification in order to develop therapeutic approaches to control ROS production and its downstream signaling pathways. Excess ROS and oxidants are caused by an imbalance between pro-oxidants and antioxidant defense systems, termed “oxidative stress” [7]. Superoxide is the parent molecule of ROS and is scavenged by an endogenous antioxidant enzyme, superoxide dismutase (Sod), converting it to hydrogen peroxide, which is further detoxified by peroxiredoxin, glutathione peroxidase, and catalase. Hence, superoxide dismutase is a first-hand defense mechanism against ROS. There are three isoforms of Sod genes, Sod 1–3, and each plays a significant role in the pathogenesis of cardiac abnormalities [8,9,10,11,12]. 

A 10-year prospective study showed that genetic variations are associated with cardiovascular disease-related and all-cause mortality in humans [13]. Superoxide dismutase 1 (Sod1) is a scavenger of superoxide anion primarily localized in the cytoplasm and mitochondrial intermembrane space. Genetic ablation of Sod1 (Sod1 knockout; Sod1KO) exacerbates infarct size and recovery of myocardial contractility after ischemia–reperfusion (IR) injury. IR injury is tissue damage caused by blood re-supply after a period of ischemia or lack of oxygen, which is associated with heightened markers of oxidative stress [12]. Likewise, transgenic mice with an increased expression of Sod1 exhibit enhanced recovery from contractile dysfunction after IR injury [8], although dose-dependent toxicity was also reported [14]. The gap in the literature, however, is the underlying morphological and functional alterations in unstressed Sod1KO hearts, which drive the impaired cardiac recovery in response to IR injury [8,12]. Sod1KO mice do not exhibit significant behavioral and physiological alterations that will significantly affect cardiac muscle health, including inactivity, decreases in food consumption, and hormonal changes [15,16].

The knockout of CuZnSOD underlies the exacerbation or protection against cardiac contractility during recovery after IR injury [8,12], but structural and functional alterations in Sod1KO hearts have not been investigated. Thus, our goal is to fill the gap in the literature, which can bridge to therapeutic strategies to protect the heart against redox-dependent abnormalities. We hypothesized that a deficiency of the cytoplasmic superoxide scavenger will lead to a functional deficit, pathological morphology, or both. We also investigated the expression and post-translational modifications of contractile proteins and potential downstream mediators activated by excess ROS in the heart. 

## 2. Results

### 2.1. Sod1KO Mice Have Elevated Oxidative Modifications in the Heart

A deficiency of CuZnSOD enzyme activity in the hearts was confirmed by activity gel (Figure 1A). We determined the protein expression of CuZnSOD using targeted mass spectrometry and confirmed its deletion (Figure 1B). Sod1KO mice exhibited ~50% elevation in lipid peroxidation products, as measured by 4-hydroxynonal (4HNE)-conjugated products in heart homogenates, compared with age-matched wildtype (WT) (Figure 1C). One of the important sources of ROS, NADPH oxidase 2 (Nox2), was found to be elevated in the Sod1KO (Figure 1D). Nox2 mRNA levels were also upregulated in the Sod1KO (Figure 1E). Together, these data exhibit markers of oxidative modifications associated with increased Nox2 enzymes in the heart. 

### 2.2. Sod1KO Mice Exhibit Increased Cardiac Systolic Function

Sod1KO has been shown to increase oxidative stress in the skeletal muscles and leads to skeletal muscle atrophy with age, along with decreased specific force [17]. To determine the effect of Sod1KO on the heart, we analyzed the cardiac systolic function by echocardiography. The analysis of cardiac function revealed increased ejection fraction and fractional shortening in the Sod1KO mice (Figure 2A,B). The internal diameter and left ventricular volume were significantly lower in the knockout mice at systole (Figure 2C), while there was a slight but significant increase in the left ventricular posterior wall thickness in the Sod1KO hearts during systole and diastole (Figure 2D). The net results of the reduced LV volume and the increased cardiac systolic function were unchanged stroke volumes. Cardiac output remained similar between WT and Sod1KO, with similar heart rates (Figure 2E).

### 2.3. Increased Heart Weights Were Manifested in Male and Female Hearts 

The normalized heart weights in the Sod1KO were significantly increased compared to WT mice. Female mice had a 19% increase and male mice exhibited a 38% increase in heart weights when normalized to body weights (Figure 3A,B), while relative organ weights from other tissues remained similar, including the kidneys and brain. (Appendix A). The heart muscle fiber cross-sectional area (CSA) was analyzed from H&E-stained histological sections. Fiber CSA in the Sod1KO hearts increased by ~30% compared to WT (Figure 3C,D).

### 2.4. Activation of Akt Signaling and Increased Ribosomal Biogenesis

Activation of the serine–threonine kinase, Akt, is one of the key downstream mediators of myocyte hypertrophy via inhibition of apoptosis, increased protein synthesis, or both. Our immunoblot analyses show increased protein abundance of total Akt and hyper-phosphorylation at Ser473 (Figure 4A), suggesting transcriptional and post-translation upregulation of Akt. Activation of Akt requires the phosphorylation of the serine 473 site [18]. The eukaryotic translation initiation factor 4E-binding protein 1, 4EBP1, is a downstream of Akt via phosphorylation. The total protein expression and phosphorylation at Thr37/46 of 4EBP1 remained unchanged in Sod1KO hearts (Figure 5C), although ROS has been shown to activate the protein in cardiomyocytes [19]. The ribosomal protein synthesis marker, S6, was hyper-phosphorylated (Figure 4B), suggesting increased protein synthesis in Sod1KO hearts. 

### 2.5. ERK and STAT3 Pathways Were Unchanged in Sod1KO Heart

ROS has been implicated as a signaling molecule that activates several transcription factors and kinases. STAT3 is a transcription factor activated by phosphorylation, which has been shown to be elevated in pathological conditions with ROS [20]. Tyr705 is one of the key sites of phosphorylation for the STAT3 activation [20,21]. Total STAT3 and phosphorylation at Tyr705 were unchanged in Sod1KO hearts (Figure 5A). Extracellular signal-regulated kinase, ERK, is also a target of ROS involved in cardiac hypertrophy, where the Thr202/204 sites are phosphorylated upon activation. Our immunoblot results exhibit no evidence of difference between Sod1KO and WT hearts (Figure 5B). 

### 2.6. Lack of Pathological Hypertrophy and Fibrotic Response in the Sod1KO Hearts

Sustained and prolonged stress or injury (pressure and volume overload) leads to decompensated hypertrophy and dilated cardiomyopathy with excessive fibrosis, cardiac dysfunction, and maladaptive gene and protein expression, including an increase in CaMKII activity, TGF-β1, and collagen 1 expression [22]. The expression of total CaMKII or oxidized (activated) CaMKII was unchanged in the hearts of the Sod1KO mice (Figure 6A), as was the expression of calcineurin (Figure 6B). We observed no increase in collagen 1 content or in the expression of the pro-fibrotic growth factor TGF-β1 in the hearts of the Sod1KO mice (Figure 6C). Real-time PCR also showed no significant change in collagen 1 or TGF-β1 mRNA expression in the hearts (Appendix A).

### 2.7. Expression and Post-Translational Modifications of the Heart Contractile Proteins Associated with Cardiac Systolic Function 

The heart contractile proteins involved in calcium-medicated force modulations were determined using targeted mass spectrometry. Vinculin was upregulated by ~30% in the Sod1KO hearts, while the rest of the contractile proteins we tested remained unchanged (Figure 7A). Phosphorylation of cardiac troponin I (cTnI) increased by ~80% in Sod1KO hearts (Figure 7B). Increased vinculin expression and cTnI phosphorylation may be associated with the increased systolic function in Sod1KO hearts. We further determined the expression of proteins and enzymes involved in β-oxidation, Krebs cycle, and carbohydrates, in which several enzymes were upregulated (Appendix A). The mitochondrial superoxide scavenger Sod2 remained unchanged, but several other antioxidant enzymes increased in Sod1KO hearts (Appendix A), as shown in other tissues challenged by redox imbalance [23,24].

## 3. Discussion

The key findings of our study are as follows: (1) a deficiency of the Sod1 enzyme, with presumably increased superoxide in the cytoplasm, induces concentric hypertrophy in mouse hearts; (2) compensatory to oxidative stress in Sod1KO, cardiac systolic function increases; and (3) the improved systolic function is associated with increased post-translational modification of TnI and increased expression of vinculin in Sod1KO.

The increased heart weights in the Sod1KO mice were accompanied by increased LV wall thickness without dilatation, which led to a significant reduction in the LV chamber diameter and LV volume (Figure 2C,D). An enhanced ejection fraction was compensatory for the smaller LV volume, resulting in no change in stroke volume or cardiac output. Cardiac wall thickening with the enhanced systolic function of the heart was consistent with a previous report showing increased ejection fraction elicited by β-adrenergic receptor activation in rats [25]. We speculate that this compensatory response is attributed to low-to-intermediate grade oxidative modifications, recently referred to as “oxidative eustress” [26]. Sod1KO hearts further challenged with ROS via IR injury exhibited a delayed recovery of LV contractility after ischemia [12], while the overexpression of Sod1 reduced infarct size and ameliorated cardiac dysfunction after IR injury [8]. It is possible that Sod1KO mice will develop cardiac dysfunction with age. Age-dependent phenotypic manifestation has been reported in this model [27]. It is interesting to note that deficiency of MnSOD, a superoxide scavenger expressed in mitochondria, has a greater impact on cardiac dysfunction and pathology compared with CnZnSOD. While whole body deletion of MnSOD leads to neonatal lethality [28,29], mice lacking myocyte-specific deletion of MnSOD demonstrated a greater level of oxidative stress with a severe dilated cardiomyopathy, with a short lifespan of ~6–8 months [9]. 

One of the signaling pathways involved in cardiac hypertrophy is Akt and its downstream pathways. Sod1KO hearts exhibited transcriptional upregulation and hyper-phosphorylation of Akt. Adaptive cardiac hypertrophy in response to exercise training has been abrogated in mice lacking Akt1 [30], demonstrating the role of Akt1 in cardiac hypertrophy. Phosphorylation of Akt occurs at several sites, which is the key to its activation. Ser473 phosphorylation induces the full activation of Akt [31], which was elevated in Sod1KO hearts. Sustained Akt activation induces pathological cardiac hypertrophy associated with mitochondrial dysfunction, which potentially leads to heart failure. Akt activates mTOR, ribosomal biogenesis, and protein synthesis [32]. Phosphorylation of ribosomal S6 in Sod1KO hearts suggests the upregulation of protein synthesis. McMullen et al. [33] demonstrated that the inhibition of mTOR by rapamycin attenuated compensatory cardiac hypertrophy elicited by aortic banding, along with the downregulation of the phosphorylation of the ribosomal S6 protein 

Redox signaling is activated by excess ROS and oxidants, which are derived from multiple intracellular sources. Nox activation is elevated in experimental models of LVH and end-stage human heart failure [34,35,36,37]. Among the five Nox isoenzymes, Nox2 and Nox4 are expressed in the heart, of which transcription and translation of Nox2 were upregulated in Sod1KO hearts. Although it is unclear how Sod1 deficiency may increase expression of Nox2, Nox2 may link the Akt activation and LV hypertrophy in Sod1KO. In support, Hingtgen et al. [35] demonstrated angiotensin II-induced Akt activation and cardiac hypertrophy, which was abrogated by adenoviral expression of Sod1. Excess ROS is also implicated in linking hyper-activity of CaMKII and STAT3 to pathological cardiac hypertrophy [34,36], but Sod1KO hearts exhibited no such evidence. Cardiac fibrosis is also reported to be manifested in oxidative stress, where Nox4 is shown to be involved [37]. Sod1KO mice failed to exhibit upregulation of Nox4, or TGF-β signaling and fibrosis. It is possible that the level of oxidative stress in Sod1KO hearts may not be sufficient to activate these pathways. 

The increased systolic function can be achieved by the altered expression of contractile proteins, post-translation modifications, or both. Our targeted mass spectrometry analysis demonstrated no evidence of difference between WT and Sod1KO hearts, except a moderate upregulation of vinculin. Vinculin is a membrane-bound protein that links the actin cytoskeleton to the sarcolemma. Highly expressed in intercalated disks and costameres, vinculin is important for force transmission and potentially force generation [38]. Cardiac-specific ablation of vinculin exhibits pathological abnormalities and may lead to ventricular tachycardia [39]. It requires further investigation whether ~30% upregulation of vinculin plays a role in compensatory hypertrophy in Sod1KO. While the rest of the contractile protein expression in the Sod1KO hearts remained similar, phosphorylation of cTnI was significantly increased at the Ser23 and 24 sites. Phosphorylation of cTnI at S23/34 increased myofibrillar calcium sensitivity and enhanced cross-bridge cycling, contributing to an increased rate of relaxation and decreased twitch duration [40]. The enhanced activation and relaxation kinetics by cTnI phosphorylation might be associated with its upstream kinase, including protein kinase A [41]. 

Contrary to the heart exhibiting wall thickening and enhanced systolic function, a deficiency of CuZnSOD elicited neurogenic atrophy and loss of specific force (force per cross-sectional area) in skeletal muscle [17]. The discrepancy between the two striated muscles might have resulted from a greater resistance or adaptation to oxidative modifications of the heart compared with skeletal muscle. Mitochondrial contents in the heart are greater than skeletal muscle. Because mitochondria are one of the primary sources of ROS, cardiac myocytes might be exposed to higher ROS than skeletal muscle in healthy states, hence rendering adaptations to oxidative modifications. In support of this, endogenous antioxidant enzymes were several folds higher in the heart than skeletal muscle [42]. The mechanisms of increased heart weights might have involved oxidants (i.e., hydrogen peroxide)-driven cardiac hypertrophy pathways. Whole body deletion of an endogenous enzyme (CuZnSOD) resulted in oxidative stress in skeletal muscle, but oxidative eustress in the heart. Note that some regulatory proteins involved in calcium-mediated force modulations are unique for cardiac muscle. Troponin I, troponin C, and troponin T are specific to the heart and have different chemical and immunochemical properties. Cardiac troponin I and troponin C also have different amino acid sequences from their skeletal muscle counterparts [43]. 

It is noteworthy that Sod1KO is a whole body knockout model, so the phenotypes exhibited in Sod1KO hearts may not be solely derived from excess ROS in hearts. A heart-specific deletion of Sod1 would be a more specific model for future investigations. Assessment of the cardiomyocyte cross-sectional area is an important parameter to evaluate cardiac hypertrophy, but the three-dimensional properties of the heart were a challenge to compare fiber sizes from identical angles. We thus used fibers with central nuclei to overcome this limitation, based on a published approach [44]. We examined the phosphorylation status of TnI, but other proteins can be phosphorylated and contribute to the enhanced systolic function of the heart. For example, hyper-phosphorylation of MyBP-C enhances the contractility of the heart [45]. Other types of post-translational modifications (i.e., acetylation) might also play a role in Sod1KO hearts. 

In conclusion, mice lacking CuZnSOD exhibited increased heart weights associated with the activation of Akt and ribosomal protein synthesis. Ejection fraction was enhanced in response to excess ROS in Sod1KO. Increased systolic function was associated with the increased protein expression of vinculin and the hyper-phosphorylation of TnI. The increased heart weight and function were driven by elevated ROS in the heart, inducing adaptation via oxidative eustress rather than oxidative stress. Finally, Sod1KO may serve as a useful tool to investigate mechanisms modulated by redox imbalance in the heart, which might contribute to cardiac abnormalities in diseases. 

## 4. Materials and Methods

### 4.1. Animal Care and Sod1^−/−^ Model

Male mice, aged 6–8 months old (average age ~7 months) were used for the study. Female C57Bl6 mice at 6–8 months old were also used for hypertrophy assessment in Figure 3A. All mice were housed in pathogen-free conditions and provided with water and food ad libitum. The Institutional Animal Care and Use Committee at the Oklahoma Medical Research Foundation approved all procedures. Sod1^−/−^ mice were generated as previously described [46]. 

### 4.2. Activity Gel 

To determine the activities of MnSOD and CuZnSOD enzymes, we homogenized ~10–20 mg of frozen muscle tissue using a buffer, containing 20 mM Tris, 0.05% Triton X, and protease inhibitor. We spun the homogenate at 14,000 rpm for 10 min and removed the supernatant. Extracts containing a portion of tissues were separated on a 10% native gel (150 V in a cold room for 2.5 h). Each gel was soaked in a solution containing nitroblue tetrazolium (NBT), riboflavin, and TEMED. The principle of this assay is based on the ability of superoxide to interact with NBT, reducing the yellow tetrazolium to a blue precipitate. Stained native activity gels show a light to dark purple appearance, with clear bands representing the area where SOD enzymes are present. Gel images were scanned in grayscale using a Syngene G Box. The upper bands represented MnSOD and the lower bands, CuZnSOD [47]. 

### 4.3. Echocardiography

Ultrasound scans of the hearts were taken using a Vevo 2100 system with an MS550D transducer (FUJIFILM VisualSonics, Toronto, ON, Canada). The mice were anesthetized with isoflurane inhalation (2–3%) for echocardiography. Measurements were taken when the heart rates were within 400–500 beats per minute and the respiration rates were around 50–100 breaths per minute. Systolic heart function was measured from the parasternal short-axis view of the heart, using echocardiography. 

### 4.4. Immunohistochemistry

The mice were euthanized at about 6–8 months of age and perfused with phosphate-based solution (PBS). The hearts were excised and embedded in 4% paraformaldehyde after sequential petrification in 15 and 30% sucrose. Heart sections at 5 µm thickness were used for histological staining. The sections were deparaffinized, rehydrated, and stained with hematoxylin and eosin (H&E). The images were analyzed using ImageJ NIH software for the quantification of positive staining. 

### 4.5. Immunoblotting

Western blots were performed using an SDS-PAGE electrophoresis system. Heart tissue lysates were prepared in a buffer containing 50 mM Tris-Cl (pH 7.4), 1 mM EDTA, 0.5 mM EGTA, 1% Triton X-100, 0.1% sodium deoxycholate, and 0.1% SDS, 140 mM NaCl. Equal amounts (10–20 µg) of protein samples were loaded onto 10–12.5% SDS PAGE gels (casting system from Bio-Rad) with 1× Tris/Glycine/SDS buffer. Proteins were transferred onto PVDF or nitrocellulose membranes overnight at 4 °C with wet-transfer equipment. Signals were detected using horse radish peroxidase (HRP)-conjugated secondary antibodies and luminol, and p-coumaric acid (PCA) for chemiluminescent detection, or using fluorescent secondary antibodies. GAPDH or total proteins using Ponceau stain were used to normalize the total protein loading. Table 1A lists the primary antibodies used in this study for Western blotting.

### 4.6. Real-Time PCR

Total RNA was extracted from the WT and Sod1KO mouse hearts using Tri Reagent (AM9738, Thermo Fisher, MA, USA) following the manufacturer’s protocol. cDNA was prepared from RNA using an iScript cDNA synthesis kit (1708891, Bio-Rad, CA, USA) and DNase 1 (18068015, Thermo Fisher) following the manufacturers’ protocols. Real-time PCR was performed using an Applied Biosystems QuantStudio 6 Flex system. The thermocycler conditions used were 50 °C for 2 min, 95 °C for 10 min, followed by 95 °C for 15 s, and 60 °C for 60 s, for a total of 40 cycles. The data were normalized to ribosomal gene 18S. Table 1B lists the real-time PCR primers used in this study.

### 4.7. Mass Spectrometry-Based Protein Analysis

High-Resolution Accurate Mass spectrometry (HRAM) analysis was used to determine the absolute concentrations of targeted proteins, as reported [48]. We loaded 20 μg of total heart homogenate samples (*n* = 3–6 biological replicates/group) on 12.5% SDS-polyacrylamide gel (Criterion, Bio-Rad). We fixed the gel and stained it with GelCode Blue (Pierce). Protein lanes were cut into ~1 mm^3^ pieces. The samples were washed, reduced with dithiothreitol (DTT), alkylated with iodoacetamide, and digested with trypsin. We extracted peptides with 70% methanol plus 5% acetic acid in water. The extracts were dried and reconstituted in 1% acetic acid. The samples were analyzed using high-resolution mass spectrometry, scanning *m*/*z* 300–1100 with a resolution of 280,000, with an orbitrap mass spectrometer (Thermo Scientific, QEx Plus) configured with a capillary column HPLC system (Thermo Scientific Ultimate 3000). The HPLC column had a ~10 cm × 75-μm inner diameter and was packed in our laboratory in a New Objective Picofrit tip with a 10-μm tip opening. The packing material was Phenomenex Aeris 3.6-μm Peptide XB-C18 100A. There were 5 μL injected. The loading phase transferred the sample from the injection loop to the column at 1.25 μL/min for 10 min. The column was eluted with the following gradients at 150 nL/min: 2-min linear gradient to 2% B, 60-min linear gradient to 45% B, 5-min linear gradient to 85% B. Solvent A was water with 0.1% (*v*/*v*) formic acid. Solvent B was 80% acetonitrile plus 20% water (*v*/*v*) with 0.1% (*v*/*v*) formic acid. The data were processed using Skyline version 3.7.0.10940 [49]. Protein abundance was determined by normalization to bovine serum albumin (BSA) used as a non-endogenous internal standard. Housekeeping proteins were also used for normalization. 

### 4.8. Statistical Analyses

Two-tailed unpaired Student *t*-tests were used to compare the means between the two groups. To test the normal distribution of the data, we performed the D’Agostino–Pearson omnibus normality test. We declared statistical significance when a *p*-value was less than 0.05. Welch’s correction was used to compare the two groups with different sample sizes, using GraphPad Prism 7.0. All data are presented as mean ± SEM.

## Figures and Tables

**Figure 1 ijms-22-02039-f001:**
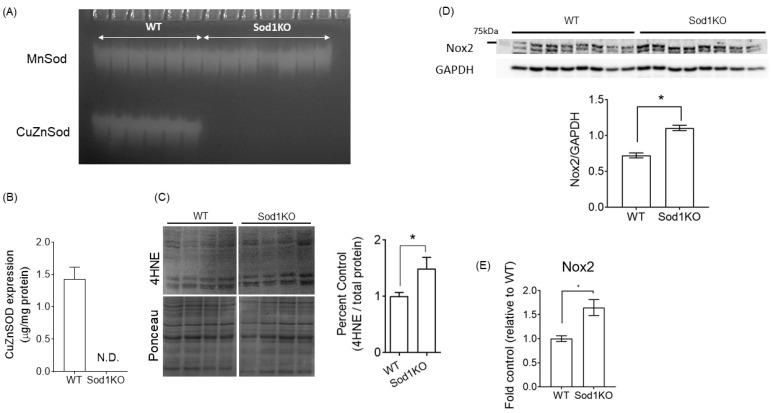
Elevated oxidative modifications in mouse hearts lacking superoxide dismutase1, Sod1 knockout or Sod1KO. (**A**) Activity gel for CuZnSOD enzyme in heart lysates from WT and Sod1KO mice (*n* = 6–8). (**B**) CuZnSOD protein contents determined by targeted mass spectrometry (*n* = 7–8). (**C**) Top: Representative blot images of molecular weights 150–40 kDa. Bottom: Quantification of the 4HNE products in heart lysates (*n* = 7–8). Entire lanes were quantified to determine the 4HNE-conjugated proteins. (**D**) Expression of Nox2 proteins and (**E**) mRNA expression, determined by qRT-PCR in the heart lysates of WT and Sod1KO mice (*n* = 8). Data are mean ± SEM. * *p* < 0.05. Abbreviations: Sod = superoxide dismutase; Nox = NADPH oxidase; 4HNE = 4-Hydroxynonenal.

**Figure 2 ijms-22-02039-f002:**
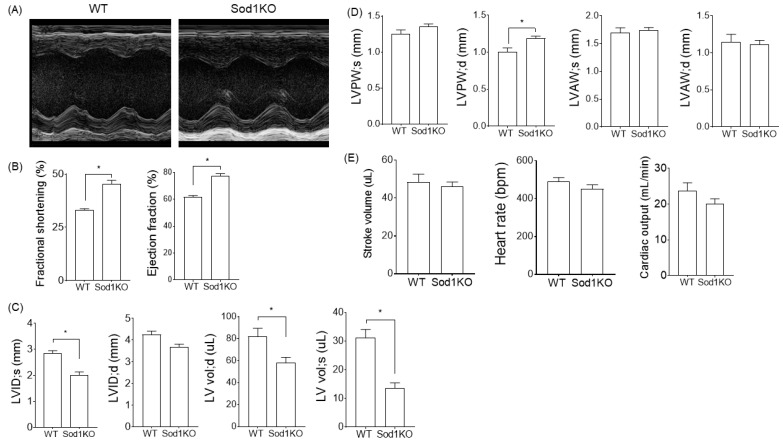
Sod1KO mice exhibited increased systolic function with compensatory remodeling. 6–8-month-old male mice were anesthetized during the assessment. Heart rates were maintained between 400–500 bpm. (**A**) Representative M-Mode images of the short axis view of WT and Sod1KO hearts. (**B**) Ejection fraction and fractional shortening in WT and Sod1KO mice. (**C**) Left ventricular internal diameter (LVID) and left ventricular volume (LV Vol) at peak systole and diastole in WT and Sod1KO mice. (**D**) Left ventricular posterior wall thickness (LVPW) at peak systole and diastole in WT and Sod1KO hearts. (**E**) Heart rate, stroke volume, and cardiac output in Sod1KO remained similar to wildtype peers. Abbreviations: LV = left ventricle; s = systole; d = diastole; LVID = LV internal diameter; LVAW = LV anterior wall thickness; LVPW = LV posterior wall thickness. *n* = 5–8. Data are mean ± SEM. * *p* < 0.05.

**Figure 3 ijms-22-02039-f003:**
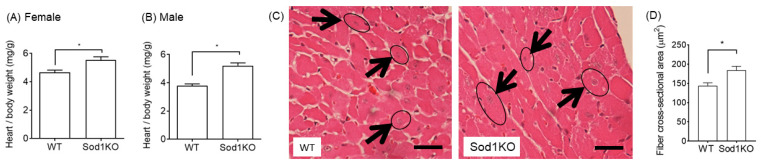
Increase in heart weights in Sod1KO mice. (**A**,**B**) Heart weights normalized to body weight significantly increased for both male and female mice (*n* = 6–8). (**C**) H&E-stained representative cross-section images of hearts from WT and Sod1KO mice. Fibers with central nuclei were analyzed for equal comparisons for fiber cross-sectional area (CSA). Arrows indicate fibers with central nuclei. The scale bar indicates 50 µm. (**D**) Myocyte CSAs were calculated from circumferences of myocytes with round and central nuclei (*n* = 4). We analyzed 15–25 fibers from each mouse to determine the mean fiber CSA from each group. * *p* < 0.05. Data are mean ± SEM.

**Figure 4 ijms-22-02039-f004:**
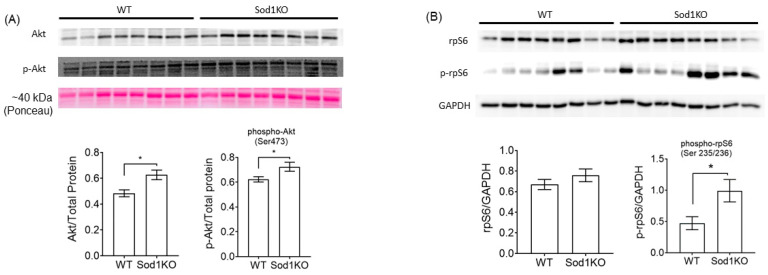
Sod1KO hearts exhibited the upregulation of Akt and ribosomal protein synthesis. (**A**) Top: Immunoblot images for total Akt and phosphorylated Akt. Bottom: Quantified data for total Akt and phosphorylated at Ser473, normalized by total proteins, were increased in Sod1KO hearts (*n* = 8). (**B**) Top: Immunoblot images for total rpS6 and phosphorylated rpS6. Bottom: quantified data for rpS6 and phosphorylated rpS6 at Ser235/236 normalized by GADPH (*n* = 8). * *p* < 0.05. Data are mean ± SEM. Abbreviations: Akt = serine–threonine kinase; rpS6 = ribosomal protein synthesis S6.

**Figure 5 ijms-22-02039-f005:**
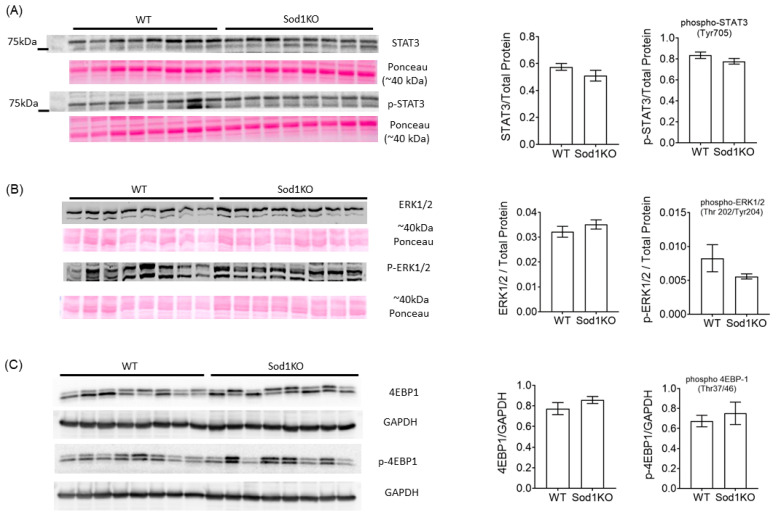
Downstream targets of oxidative modifications remained unchanged in Sod1KO. (**A**) Left: Immunoblot images showing total STAT3 and phosphorylated STAT3 in heart lysates. Right: Quantified total STAT3 and phosphorylated STAT3 at Tyr705, normalized to total proteins (*n* = 8). (**B**) Left: Immunoblot images showing total ERK1/2 and phosphorylated ERK1/2. Right: Quantified total ERK1/2 and phosphorylated ERK1/2 at Thr202 and Tyr204, normalized to total proteins (*n* = 8). (**C**) Top: Immunoblot images showing total 4EBP1 and phosphorylated 4EBP1 in heart lysates. Bottom: Quantified total 4EBP1 and phosphorylated 4EBP1 at Thr37 and 46 sites, normalized to GAPDH (*n* = 8). Data are mean ± SEM. Abbreviations: 4EBP1 = eukaryotic translation initiation factor 4E-binding protein 1; ERK = extracellular signal-regulated kinases.

**Figure 6 ijms-22-02039-f006:**
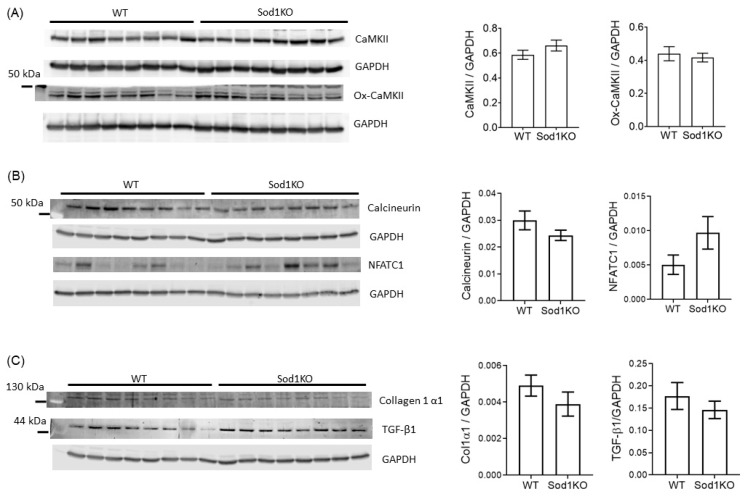
Lack of hypertrophic and fibrotic response in Sod1KO hearts. (**A**) Left: Immunoblot images showing total CaMKII and oxidized CaMKII. Right: Quantified total CaMKII and oxidized CaMKII (ox-CaMKII), normalized to GAPDH (*n* = 8). (**B**) Left: Immunoblot images showing calcineurin and NFATc1. Right: Quantified optical densities for calcineurin and NFATc1, normalized to GAPDH (*n* = 8). (**C**) Left: Immunoblot images showing collagen1α1 and TGF-β1. Right: Quantified optical densities for collagen1α1 and TGF-β1, normalized to GAPDH (*n* = 8). Data are mean ± SEM. Abbreviations: CaMKII = Ca^2+^/calmodulin-dependent protein kinase II; NFATC1 = nuclear factor of activated T-cells cytoplasmic; Col1a1 = collagen type I alpha 1 chain; TGF-β1 = transforming growth factor-β1.

**Figure 7 ijms-22-02039-f007:**
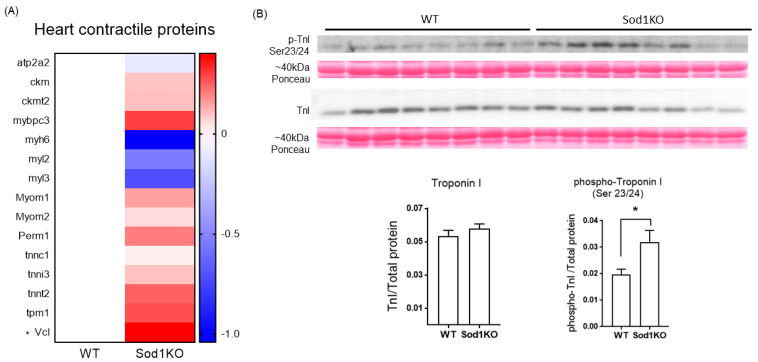
Heart contractile protein abundance and troponin I phosphorylation. (**A**) Heart contractile proteins determined by targeted mass spectrometry (*n* = 8). (**B**) Top: Immunoblot images showing total troponin I (cTnI) and phosphorylated cTnI in heart lysates. Bottom: Quantified total cTnI and phosphorylated cTnI at Ser23 and 24 sites, normalized to total proteins (*n* = 8). Data are mean ± SEM. * *p* < 0.05. Abbreviations: cTnI = cardiac troponin I.

**Table 1 ijms-22-02039-t001:** Sources of antibodies and primers used for the study.

**(A) List of Primary Antibodies Used for Immunoblots**					
**Primary Antibody**		**Catalog Number**	**Source**		**Host**
4HNE		393207	EMD Millipore		Rabbit
Nox2		Ab80508	Abcam		Rabbit
Akt		4691	Cell Signaling Technologies		Rabbit
p-Akt		4060	Cell Signaling Technologies		Rabbit
ERK1/2		9102	Cell Signaling Technologies		Rabbit
p-ERK1/2		4370	Cell Signaling Technologies		Rabbit
STAT3		4904	Cell Signaling Technologies		Rabbit
p-STAT3		9145	Cell Signaling Technologies		Rabbit
rpS6		2217	Cell Signaling Technologies		Rabbit
p-rpS6		SC-293144	Santa Cruz Biotechnology		Mouse
4EBP1		9644	Cell Signaling Technologies		Rabbit
p-4EBP1		2855	Cell Signaling Technologies		Rabbit
GAPDH		G9545	Sigma		Rabbit
CaMKII		Ab22609	Abcam		Mouse
ox-CaMKII		07-1387	EMD Millipore		Rabbit
Calcineurin A		2614	Cell Signaling Technologies		Rabbit
NFATC1		SC-7294	Santa Cruz Biotechnology		Mouse
Collagen 1 α1		Ab34710	Abcam		Rabbit
TGF-β1		AF-101-NA	R&D systems		Chicken
**(B) List of Mouse Primers Used for qRT PCR**					
**Gene**	**Forward Primer**			**Reverse Primer**	
Tgf-b1	CAA GGG CTA CCA TGC CAA CT			GTA CTG TGT GTC CAG GCT CCA A	
Col1a1	GAA GCA CGT CTG GTT TGG A			ACT CGA ACG GGA ATC CAT C	

## Data Availability

The data that support the findings of this study are available from the corresponding author upon reasonable request.

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
