# Peer review of "Myocardial Hypertrophy and Compensatory Increase in Systolic Function in a Mouse Model of Oxidative Stress"

_ijms, 2021, doi:10.3390/ijms22042039_

Round 1

Reviewer 1 Report

I read with interest the paper by Varshney et al regarding the myocardial functional, histological and physiological alteration in SOD1KO mice. The authors conclude that mice lacking cy27 toplasmic superoxide dismutase are associated with an increase in heart weight and concentric hy28 pertrophy, exhibiting a pathological adaptation of the heart to oxidative stress. Their findings are well supported by their data, yet there are minor edits as follows:

-Please proof read the whole manuscript to revise minor errors in grammar and syntax.

-The introduction section is a bit wordy and should be truncated.

-Was any other method used to assess cardiac function such as intracardiac pressure measurement, PV loops etc?

-How do the authors justify this wide range in the age of the mice. Were there any differences between the groups?

-The authors state that t-test was used. How was normality assessed and confirmed for the distribution of their data?

Author Response

I read with interest the paper by Varshney et al regarding the myocardial functional, histological and physiological alteration in SOD1KO mice. The authors conclude that mice lacking cytoplasmic superoxide dismutase are associated with an increase in heart weight and concentric hypertrophy, exhibiting a pathological adaptation of the heart to oxidative stress. Their findings are well supported by their data, yet there are minor edits as follows:

Thank you for your interest in our manuscript.

-Please proof read the whole manuscript to revise minor errors in grammar and syntax.

We modified the text throughout the manuscript.

-The introduction section is a bit wordy and should be truncated.

The introduction of the manuscript is less than a page with key information leading to the reason for Sod1KO heart investigation. We think that the current contents are necessary for readers who are not redox biologists.

-Was any other method used to assess cardiac function such as intracardiac pressure measurement, PV loops etc?

We performed echocardiography to measure cardiac function in vivo. The intracardiac pressure assessment and PV loop would be great additions to our work in the future. Thank you for your suggestion.

-How do the authors justify this wide range in the age of the mice. Were there any differences between the groups?

This is a good point. We removed all the data from 9-11 months old mice. Now, all the data in the manuscript include 6-8 months old mouse heart. We extensively modified our text in Figure 2.

-The authors state that t-test was used. How was normality assessed and confirmed for the distribution of their data?

This is great point. We modified our text as follows in the statistical analysis section as follows. To test normal distribution of the data, we performed D’Agostino-Perarson omnibus normality test as recommended by GraphPad Prism 7.0.

Reviewer 2 Report

In the current manuscript, the authors characterized the morphological and functional changes in unstressed Sod1 knockout hearts. The authors found that Sod1 knockout mice developed concentric cardiac hypertrophy with enhanced cardiac systolic function. But the authors failed to identify the underlying mechanism. The data quality is low, the experimental design was not clear, which largely impaired the impacts of this study. 

Points:

  1. The main contribution of the current study is that Sod1 KO mice developed concentric cardiac hypertrophy with enhanced cardiac systolic function. But the authors did not give clear descriptions of the experimental time points. The authors only claimed that mice used in this study were aged 6-11 months old in the Materials & Methods section, which makes it extremely difficult to interpret these data. 
    1. According to the previous study (Sakellariou G et al., 2018), no significant cardiac hypertrophy was detected in the Sod1 knockout mice at ~9 months of age. So, it’s critical to know at what time point the authors identified the cardiac hypertrophy phenotype in the current study (Fig. 3A).
    2. Figure 2, what is the age of these mice? Is there any time point that these Sod1 KO mice have normal cardiac function and morphology? This information is critical for studies trying to uncover the underlying mechanism. 
    3. Also, did the authors examine the cardiac function at a late stage to see if the concentric cardiac hypertrophy will ultimately transits to dilated cardiomyopathy?
  2. Figure 3C, the representative H&E pictures gave very limited information. The authors may want to perform the Wheat Germ Agglutinin Staining, which will show a much more obvious cardiomyocyte shape. Also, how many cells were measured per mouse? It will be very helpful to give the cardiomyocyte size distribution besides the average (Fig. 3D). 
  3. Figure 4, these changes are too subtle. The western blot data varied too much, which did not reflect the quantification results. 
  4. All western blot images should include size markers bracketing the bands of interest.
  5. Also, what’s the age of these mice used in Figure 4. If the authors wanted to set up a cause and consequence relation between these molecular changes and cardiac hypertrophy, these samples should be collected at the time point before the cardiac hypertrophy developed.  
  6. Figure 6, the authors may also want to perform Sirius red staining and/or hydroxyproline assay to see if there is any fibrotic response at a late stage. 
  7. Supplemental figure 2, calcineurin, NFATC1 results were missing. 
  8. Line 336-337 “Primary antibody used was Ki-67…..”, no Ki-67 data was presented in this study.

Round 2

Reviewer 2 Report

The authors addressed my comments.